# Technical note: Optimizing the in situ cosmogenic $^{36}$Cl extraction and measurement workflow for geologic applications

Alia J. Lesnek[1,2,3], Joseph M. Licciardi[1], Alan J. Hidy[4], Tyler S. Anderson[4]

[1]Department of Earth Sciences, University of New Hampshire, Durham, NH 03824, USA
[2]School of Earth and Environmental Sciences, Queens College, CUNY, Flushing, NY 11367, USA
[3]Department of Earth and Environmental Sciences, The Graduate Center, CUNY, New York, NY 10016, USA
[4]Center for Accelerator Mass Spectrometry, Lawrence Livermore National Laboratory, Livermore, CA 94550, USA

*Correspondence to*: Alia J. Lesnek (alia.lesnek@qc.cuny.edu)

**Abstract.** In situ cosmogenic $^{36}$Cl analysis by accelerator mass spectrometry (AMS) is routinely employed to date Quaternary surfaces and assess rates of landscape evolution. However, standard laboratory preparation procedures for $^{36}$Cl dating require the addition of large amounts of isotopically enriched chlorine spike solution; these solutions are expensive and increasingly difficult to acquire from commercial sources. In addition, the typical workflow for $^{36}$Cl dating involves measuring both $^{35}$Cl/$^{37}$Cl and $^{36}$Cl/Cl concurrently on the high-energy (post-accelerator) end of the AMS system, but $^{35}$Cl/$^{37}$Cl determinations using this technique can be complicated by isotope fractionation and system memory during measurement. The traditional workflow also does not provide $^{36}$Cl extraction laboratories with the data needed to calculate native Cl concentrations in advance of $^{36}$Cl/Cl measurements. In light of these concerns, we present an improved workflow for extracting and measuring chlorine in geologic materials. Our initial step is to characterize $^{35}$Cl/$^{37}$Cl on up to ~1 g sample aliquots prepared in Ag(Cl, Br) matrices, which greatly reduces the amount of isotopically enriched spike solution required to measure native Cl content in each sample. To avoid potential issues with isotope fractionation through the accelerator, $^{35}$Cl/$^{37}$Cl is measured on the low-energy, pre-accelerator end of the AMS line. Then, for $^{36}$Cl/Cl measurements, we extract Cl as AgCl or Ag(Cl, Br) in analytical batches with a consistent total Cl load across all samples; this step is intended to minimize source memory effects during $^{36}$Cl/Cl measurements and allows for preparation of AMS standards that are customized to match known Cl contents in the samples. To assess the efficacy of this extraction and measurement workflow, we compare chlorine isotope ratio measurements on seven geologic samples prepared using standard procedures and the updated workflow. Measurements of $^{35}$Cl/$^{37}$Cl and $^{36}$Cl/Cl are consistent between the two workflows, and $^{35}$Cl/$^{37}$Cl values measured using our methods have considerably higher precision than those measured following standard protocols. The chemical preparation and measurement workflow presented here (1) reduces the amount of isotopically enriched chlorine spike used per rock sample by up to 95%, (2) identifies rocks with high native Cl concentrations, which may be lower priority for $^{36}$Cl surface exposure dating, at an early stage of analysis, and (3) allows laboratory users to maintain control over the total chlorine content within and across analytical batches. These methods can be incorporated into existing laboratory and AMS protocols for $^{36}$Cl analyses and will increase the accessibility of $^{36}$Cl dating for geologic applications.

**Short summary:** We present an improved workflow for extracting and measuring chlorine isotopes in rocks and minerals. Experiments on seven geologic samples demonstrate that our workflow provides reliable results while offering several distinct advantages over traditional methods. Most notably, our workflow reduces the amount of isotopically enriched chlorine spike used per rock sample by up to 95%, which will allow researchers to analyze more samples using their existing laboratory supplies.

## 1 Introduction

Cosmogenic $^{36}$Cl is widely used within the geosciences to determine the duration of surface exposure of Quaternary features such as glacial deposits (Barth et al., 2019; Phillips et al., 1997; Small et al., 2016), lava flows (Parmelee et al., 2015; Singer et al., 2018), landslides (Ivy-Ochs et al., 2009; Zerathe et al., 2014; Pánek et al., 2018), terraces (Kozaci et al., 2007; Robertson et al., 2019), and fault scarps (Benedetti et al., 2002; Mitchell et al., 2001; Schlagenhauf et al., 2011). Over the past few decades, $^{36}$Cl has also emerged as the primary isotope for constraining rates of landscape evolution in carbonate settings (Ben-Asher et al., 2021; Marrero et al., 2018; Stone et al., 1996). $^{36}$Cl offers several advantages over other commonly used cosmogenic isotopes (e.g., $^{10}$Be) for surficial geochronology. $^{36}$Cl is produced by multiple reactions in a wide variety of minerals, including orthoclase, plagioclase, and calcite (Gosse and Phillips, 2001), enabling its use in dating mineral separates and whole-rock samples of nearly any lithology. The high production rate of $^{36}$Cl (Marrero et al., 2016) also allows for age determinations on young materials (e.g., Price et al., 2022). Additionally, accelerator mass spectrometry (AMS) measurements of chlorine (prepared as AgCl) have low detection limits and high beam currents (Finkel et al., 2013). Finally, recent advances in $^{36}$Cl production rate calibrations (Marrero et al., 2016) and the availability of web-based calculators (e.g., CRONUScalc, cronus.cosmogenicnuclides.rocks/2.1/html/cl/; CRONUSEarth, stoneage.ice-d.org/math/Cl36/v3/v3_Cl36_age_in.html; CREP, https://crep-dev.otelo.univ-lorraine.fr/#/init) have enabled surface exposure ages or erosion rates to be determined with relative ease once total sample Cl, $^{36}$Cl concentration, and elemental concentrations are obtained.

In situ $^{36}$Cl concentrations are typically measured via AMS methods on targets prepared in an AgCl matrix (Fig. 1; Licciardi et al., 2008). To ensure that Cl isotope ratios are well above laboratory blank values, consistent sample masses are prepared for Cl isotope analysis; depending on anticipated $^{36}$Cl inventories (which are a function of exposure duration, altitude, and sample composition), each sample usually consists of ~10-20 g of milled rock for whole-rock silicates or ~5-10 g of isolated mineral separates. Rock samples are spiked with isotopically enriched Cl carrier solution such that total sample Cl (from $^{35}$Cl/$^{37}$Cl) and $^{36}$Cl concentrations (from $^{36}$Cl/$^{37}$Cl or $^{36}$Cl/Cl) can be determined through isotope dilution methods (Faure and Mensing, 2005). At the University of New Hampshire and the Center for AMS (CAMS) at Lawrence Livermore National Laboratory (LLNL), geologic samples have historically been spiked with ~750-1000 μg of Cl from a $^{37}$Cl-enriched solution with a $^{35}$Cl/$^{37}$Cl of approximately 1, which is substantially lower than the natural $^{35}$Cl/$^{37}$Cl of 3.127. Following standard protocols (Licciardi et al., 2008), Cl is extracted from the samples as AgCl. After shipment to CAMS, the AgCl precipitates are packed into AgBr plugs that are pressed into open-faced stainless steel cathodes. Both $^{35}$Cl/$^{37}$Cl and $^{36}$Cl/Cl are then

measured simultaneously on the high-energy (i.e., post-accelerator) end of the 10 MeV Tandem Van de Graaff accelerator at CAMS, and isotope extraction laboratories receive $^{35}$Cl/$^{37}$Cl and $^{36}$Cl/Cl results after all measurements are completed.

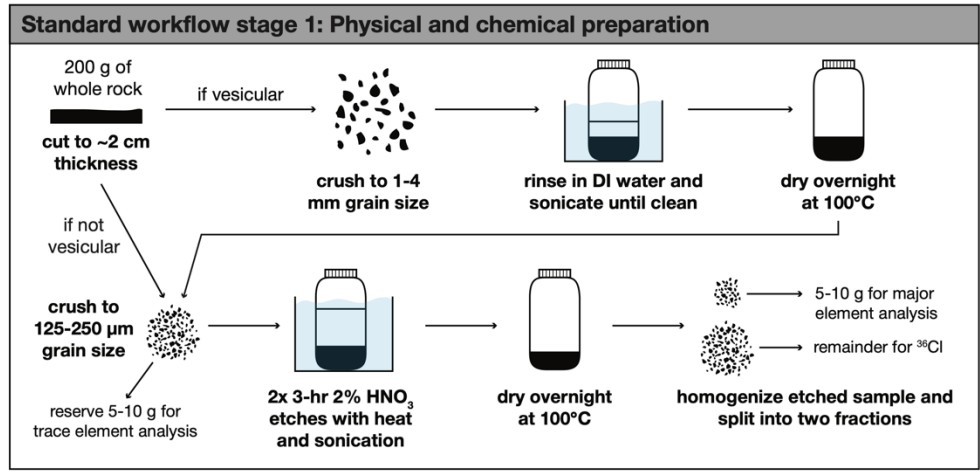

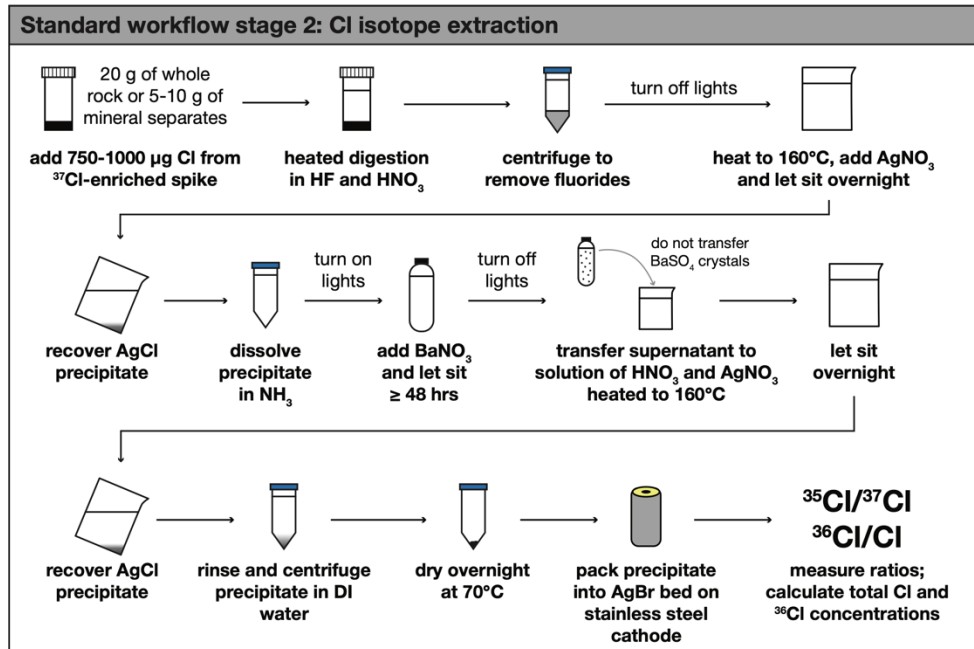

**Figure 1: Schematic of the standard workflow for cosmogenic $^{36}$Cl analysis based on Licciardi et al. (2008). Black arrows indicate the order of steps for each stage of the process. Note that while we use a $^{37}$Cl-enriched spike solution for the stable Cl extraction, these procedures are also suitable for use with a $^{35}$Cl-enriched spike.**

While this workflow has proven useful over many years of Cl measurements at CAMS, there are several areas in which procedural and performance improvements are possible. First, few commercial sources of $^{37}$Cl-enriched solutions exist

with negligible $^{36}Cl$ concentrations that are acceptable as carrier for $^{36}Cl$ sample preparation, and available source materials are extremely expensive. Alternatively, commercial sources of $^{35}Cl$-enriched carrier are more readily obtained and available with negligible $^{36}Cl$ concentrations. However, when $^{36}Cl$, $^{35}Cl$, and $^{37}Cl$ are measured simultaneously at the high energy end of an AMS system, it is undesirable to use carrier enriched in the more abundant isotope ($^{35}Cl$). This is because it inflates the difference in intensity between $^{35}Cl^-$ and $^{37}Cl^-$ beams injected into the accelerator, leading to increased instability in maintaining terminal voltage. Considering these issues, the supply of $^{37}Cl$-enriched spike that is currently being used for sample preparation in extraction laboratories is limited and/or undesirable unless laboratory users have the means to invest substantial sums for new isotopically enriched chlorine solutions. Second, in the sample preparation methods outlined above, the native Cl content in each geologic sample is not known in advance of the $^{36}Cl/Cl$ measurements, hence it is not possible to control total Cl in an analytical batch of AMS targets. Variable total Cl among samples can cause AMS memory effects after ionization of unexpectedly high Cl targets (Arnold et al., 2010; Finkel et al., 2013), and reduction of these memory effects involves longer measurement times per sample. Finally, measurements of stable Cl ratios on the high-energy, post-accelerator portion of the AMS can be affected by isotope fractionation in the terminal stripper at high (>40 µA) $^{35}Cl^-$ beam currents, leading to inaccurate determinations of $^{35}Cl/^{37}Cl$ (Wilcken et al., 2013). Accurate $^{35}Cl/^{37}Cl$ measurements are essential for determining relative contributions of different $^{36}Cl$ production pathways to the cosmogenic $^{36}Cl$ inventory in a sample, and for deriving the cosmogenic $^{36}Cl$ concentration and exposure ages.

In this technical note, we present an improved laboratory and analytical workflow for measuring Cl isotope concentrations in silicate rocks via AMS. This workflow is similar to procedures developed at the University of Washington Cosmogenic Isotope Laboratory (Stone et al., 1996; Stone, 2001), but with a key difference being that we strongly encourage laboratory users to prepare subsamples for $^{35}Cl/^{37}Cl$ measurements in advance of $^{36}Cl/Cl$ analyses. While this workflow was implemented specifically for silicates, where hydrofluoric acid (HF) is required for full digestion, it is suitable for carbonates as well by simply modifying the digestion step to exclude HF. Through a set of experiments on geologic samples, we demonstrate that the updated workflow provides comparable and, in some cases, more precise results than the standard workflow. A key finding of our experiments is that compared to standard methods, our workflow reduces the use of costly isotopically enriched Cl spike solution by up to 95%, which should increase the accessibility of $^{36}Cl$ dating for geologic applications, as laboratory users can prepare more samples with their existing supplies.

## 2 Methods

Our improved cosmogenic Cl workflow involves measuring sample chloride concentrations via isotope dilution on a small (~1 g) aliquot of rock prior to digestion of the full rock sample (Fig. 2). $^{35}Cl/^{37}Cl$ analyses are performed on the low energy (pre-accelerator) end of an AMS with as little as 50 µg of isotopically enriched Cl added to the target. This reduction in Cl is compensated by bulking the sample with bromine (from carrier made with commercially sourced $NH_4Br$) and co-precipitating as $Ag(Cl, Br)$ to facilitate consistent sample handling and fully packed targets. Once total Cl concentrations have

been determined for each sample, rock samples are prepared for $^{36}Cl/Cl$ measurements using a modified version of standard $^{36}Cl$ methods, with an optional addition of $NH_4Br$ carrier at the dissolution stage for samples with low total Cl loads. $^{36}Cl/Cl$ measurements are made on AgCl or Ag(Cl, Br) targets containing a minimum of 500 µg Cl, but ideally 750-1000 µg Cl as AgCl.

This approach offers several advantages compared to the traditional workflow. Because Cl concentrations are determined before preparing a rock sample for $^{36}Cl$ analysis, there is no need to add an isotopically enriched carrier solution to the full sample. Instead, if additional Cl is required, a carrier solution containing natural ratio Cl can be used (Stone, 2001). The natural-ratio Cl solution can be made with widely available and inexpensive material such as mined rock salt that is typically low in $^{36}Cl$. With this approach, each sample in an analytical $^{36}Cl/Cl$ measurement batch is tuned to contain a similar

amount of total chlorine, the $^{37}Cl$-enriched spike solution is conserved, and the presence of significant $^{36}Cl$ in the enriched spike solution becomes a nonissue since it is used only for the stable Cl aliquots and not the target measured for $^{36}Cl/Cl$.

      We tested this workflow on seven whole-rock geologic samples with varying silicate lithologies and a wide range of expected $^{36}Cl$ inventories. For comparison, we also measured stable Cl and cosmogenic $^{36}Cl$ on splits of the same samples prepared using standard procedures (Fig. 1; Licciardi et al., 2008). All chlorine isotope ratio measurements were performed at

120 LLNL-CAMS between July 2020 and December 2023. The setup of the CAMS accelerator and operational parameters for $^{35}Cl/^{37}Cl$ measurements are described in detail in Anderson et al. (2022).

### 2.1 Cl dilution series

      To assess the accuracy of $^{35}Cl/^{37}Cl$ measurements on the low-energy (i.e., pre-accelerator) portion of the AMS, we prepared a dilution series of four samples using varying amounts of $^{37}Cl$-enriched spike, a natural-ratio Cl carrier, and $NH_4Br$

carrier. For this dilution series, we used a $^{37}Cl$-enriched solution prepared at the University of New Hampshire ("Wildcat spike"). The Wildcat spike has a $^{35}Cl/^{37}Cl$ of 1.001 and a Cl concentration of $942 \pm 15$ µg g$^{-1}$ (measured via ICP-OES; n =4; average $\pm$ one standard deviation). To make the Wildcat spike, we combined $^{37}Cl$-enriched NaCl powder with a natural-ratio Cl carrier solution to achieve the desired $^{35}Cl/^{37}Cl$ of 1.001. The isotopically enriched NaCl powder was purchased in October 2004 from Oak Ridge National Laboratory (ORNL) and had an atomic assay of 98.21% $^{37}Cl$ and 1.79% $^{35}Cl$ (ORNL batch

198501). The natural-ratio Cl carrier solution ("Weeks Island Halite carrier") was made at the University of New Hampshire by dissolving NaCl obtained from a mine in Weeks Island, Louisiana in deionized water. The Cl concentration in the Weeks Island Halite carrier is $1436 \pm 9$ µg g$^{-1}$ (measured via ICP-OES; n = 4; average $\pm$ one standard deviation). In the dilution series, we adjusted the amounts of Wildcat Spike and Weeks Island Halite carrier such that total Cl in each sample was ~150 µg while the expected $^{35}Cl/^{37}Cl$ values ranged from 1.001 to 2.510 (Table 1). All samples were bulked with ~4000 µg Br from a 10,000

135 µg g$^{-1}$ $NH_4Br$ carrier solution.

**Table 1: Results of $^{35}Cl/^{37}Cl$ measurements of a dilution series to test low-energy AMS performance across a range of $^{35}Cl/^{37}Cl$ values expected in geologic samples.**

| Sample ID | Cl added from Wildcat spike (μg)[a] | Cl added from WIH carrier (μg)[b] | Expected $^{35}Cl/^{37}Cl$ | Measured $^{35}Cl/^{37}Cl$ | Measured $^{35}Cl/^{37}Cl$ uncertainty | Number of measurements | Average $^{37}Cl$ current (μA) | Average $^{35}Cl$ current (μA) |
|---|---|---|---|---|---|---|---|---|
| CLDSB2-1 | 150 | 0 | 1.001 | 0.997 | 0.001664 | 15 | 3.927 | 3.954 |
| CLDSB2-2 | 94 | 56 | 1.484 | 1.455 | 0.002427 | 15 | 2.955 | 4.341 |
| CLDSB2-3 | 53 | 98 | 2.016 | 1.997 | 0.003331 | 17 | 1.163 | 2.335 |
| CLDSB2-4 | 25 | 125 | 2.510 | 2.507 | 0.004182 | 16 | 0.744 | 1.875 |

[a] The "Wildcat spike" has a measured $^{35}Cl/^{37}Cl$ of 1.001 and a Cl concentration of $942 \pm 15$ μg g$^{-1}$.
[b] The "Weeks Island Halite" (WIH) carrier has a $^{35}Cl/^{37}Cl$ of 3.127 (i.e., natural ratio) and a Cl concentration of $1436 \pm 9$ μg g$^{-1}$.

To quantify the magnitude of Cl contamination in our bromine carrier, we prepared a second dilution series using the $^{37}Cl$-enriched Wildcat spike and the $NH_4Br$ carrier. We prepared five samples for this dilution series with Cl loads ranging from ~50 to 150 μg Cl. All samples received ~0.4 g of 10,000 μg g$^{-1}$ $NH_4Br$ solution (~4000 μg Br; Table 2). For both dilution series, after the carrier additions, the solutions were acidified in 8 mL of 2M $HNO_3$. To precipitate Ag(Cl, Br), ~1 mL of 5% $AgNO_3$ was added to each solution under low light conditions and left to precipitate for at least 12 hours. The precipitates were then rinsed, dried, and packed into acid-cleaned stainless steel cathodes for $^{35}Cl/^{37}Cl$ measurement at CAMS.

**Table 2: $^{35}Cl/^{37}Cl$ measurements of a dilution series to test Cl contamination in the commercial $NH_4Br$ source used in preparation of geologic test samples.**

| Sample ID | Cl added from spike (μg)[a] | Br added from carrier (μg)[b] | Memory-corrected $^{35}Cl/^{37}Cl$ | $NH_4Br$ carrier Cl concentration (μg g$^{-1}$) |
|---|---|---|---|---|
| CLDSA2-1 | 141 | 4099 | 1.009 | 2.14 |
| CLDSA2-2 | 134 | 4071 | 1.010 | 2.20 |
| CLDSA2-3 | 94 | 4101 | 1.011 | 1.76 |
| CLDSA2-4 | 71 | 4093 | 1.013 | 1.64 |
| CLDSA2-5 | 47 | 4072 | 1.012 | 1.02 |
| | | | mean = | $1.75 \pm 0.473$ |

[a] Samples were spiked with the "Wildcat Spike," which has a measured $^{35}Cl/^{37}Cl$ of 1.001 and a Cl concentration of $942 \pm 15$ μg g$^{-1}$.
[b] The carrier solution has a concentration of ~10,000 μg g$^{-1}$ $NH_4Br$.

## 2.2 Preparation of test samples

We prepared a set of seven geologic samples for $^{35}Cl/^{37}Cl$ and $^{36}Cl$ analyses (Fig. 2). Four samples (19SEAK-01, -02, -12, and -13) were collected in 2019 from two locations in coastal Southeast Alaska. 19SEAK-01 and 19SEAK-02 were sampled from erratic boulders of olivine basalt on Suemez Island (Eberlein et al., 1983) that were deposited by the Cordilleran Ice Sheet during the last deglaciation (Walcott et al., 2022). Samples 19SEAK-12 and 19SEAK-13 were obtained from the

surface of a vesicular plagioclase basalt flow that was emplaced in the eastern Mount Edgecumbe Volcanic Field sometime during the late Pleistocene (Riehle et al., 1989). Three rhyolite samples (YGT18-31, -32, and -33) were collected in 2018 from the Yellowstone Plateau. These samples were taken from the top surfaces of erratic boulders deposited during the most recent deglaciation of the Yellowstone Ice Cap (Licciardi and Pierce, 2018).

After initial physical and chemical preparation, we split each rock sample into two fractions. We prepared the "A" splits (e.g., 19SEAK-01A) using standard procedures (Fig. 1; Stone et al., 1996; Licciardi et al., 2008) where targets are prepared in an AgCl matrix and stable Cl ratios and cosmogenic $^{36}Cl$ are measured concurrently on the post-accelerator end of the AMS. We used the "B" splits (e.g., 19SEAK-01B) to test the updated workflow (Fig. 2). For these samples, we first characterized the stable Cl ratios and total sample Cl on a small aliquot removed from the full sample. Using this information,
we then prepared the rock samples for $^{36}Cl$ analysis.

### 2.2.1 Physical and chemical preparation: All test samples

        We prepared all whole rock samples at the University of New Hampshire Cosmogenic Isotope Lab. Each rock sample had a starting mass of 200-300 g and a thickness of 2-3 cm. We cut some samples with a tabletop rock saw to achieve a uniform thickness. To remove dirt and other contaminants from vesicles, we first crushed samples 19SEAK-12 and 19SEAK-13 to a
1-4 mm grain size. We rinsed these two crushed samples in deionized water and sonicated them in 10-minute intervals until no further material could be removed. After the initial crushing and rinsing, we dried the 1-4 mm fractions overnight at 100 °C, and then crushed all seven samples to a 125-250 μm grain size. At this stage, we reserved 5-10 g of the 125-250 μm size fraction for trace element analysis, which is necessary to characterize the neutron moderating properties of the rock for $^{36}Cl$ exposure age calculations. From the remainder of the 125-250 μm size fraction, we rinsed ~50 g in deionized water. We leached
the samples in a 2% HNO₃ solution for three hours in heated ultrasonic tanks, rinsed the material in deionized water, and repeated the leaching for a total of two 2% HNO₃ leaches. After the leached samples were dried overnight at 100°C, we divided the remaining material into two fractions to compare results from standard preparation methods ("A" splits; Fig. 1) and the updated workflow ("B" splits; Fig. 2). In the next sections, we describe the procedures used for the updated workflow in detail.

### 2.2.2 Characterization of stable Cl ratios: Updated workflow

The goal of this step is to characterize stable chlorine ratios ($^{35}Cl/^{37}Cl$) of geologic samples prior to $^{36}Cl$ chemistry on the full rock sample. For the test samples, we carried out this process on a ~1 g aliquot that was removed from the rock sample before beginning the $^{36}Cl$ extraction (Table 3). After the two 2% HNO₃ etches and subsequent drying at 100 °C, we divided the sample into three fractions using a small spoon (Fig. 2). The first fraction was the ~1 g aliquot for stable Cl analysis. The second fraction consisted of ~5-10 g for major element analysis, which is necessary to characterize $^{36}Cl$ production rates for
exposure dating purposes. We reserved the remaining material for $^{36}Cl$ measurements. To assess the effect of sample homogenization prior to splitting on measured total Cl concentration, we also prepared aliquots of four samples (19SEAK-01, -02, -12, and YGT18-31) using a micro riffle splitter (rather than a spoon) to separate the subsamples from the full rock.

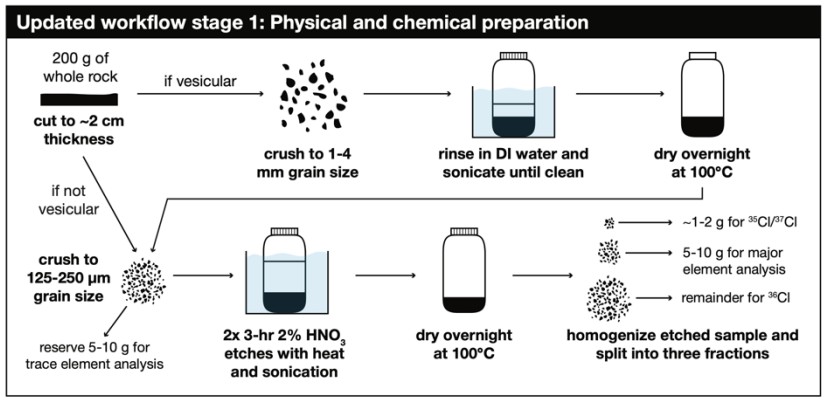

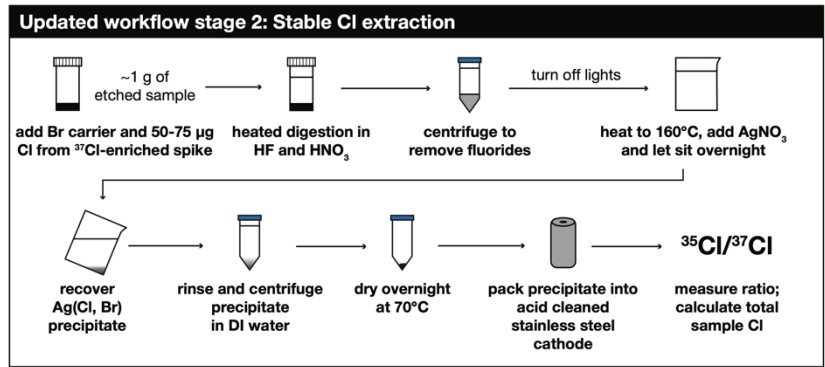

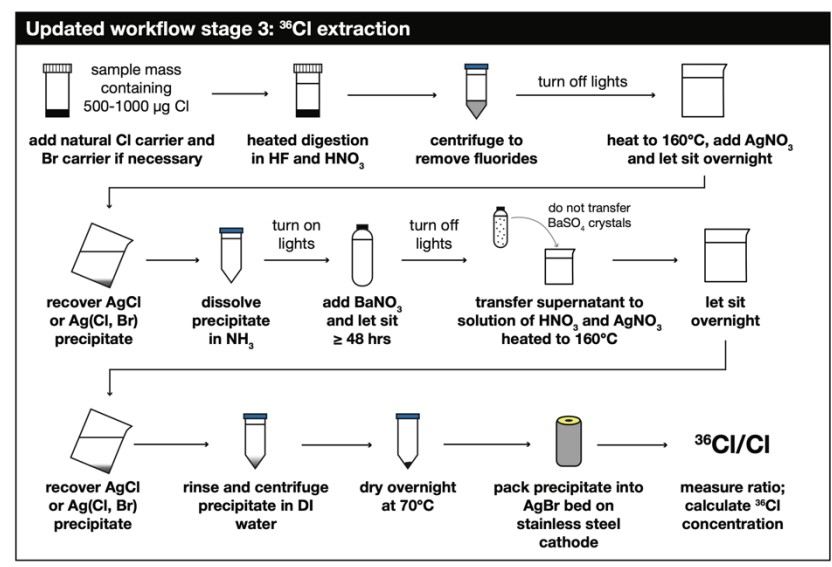

**Figure 2: Schematic of the updated workflow for cosmogenic $^{36}$Cl analysis presented here. Black arrows indicate the order of steps for each stage of the process. Note that while we use a $^{37}$Cl-enriched spike solution for the stable Cl extraction, these procedures are also suitable for use with a $^{35}$Cl-enriched spike.**

We measured $^{35}Cl/^{37}Cl$ on Ag(Cl, Br) targets prepared using the ~1 g aliquots of etched rock sample. To prepare the Ag(Cl, Br) targets, we spiked the aliquots with a small amount of LLNL Spike A (Cl concentration = 1284 ± 4 µg g$^{-1}$, $^{35}Cl/^{37}Cl$

= 0.934), totalling ~75 µg of Cl. We then added ~4000 µg of bromine to each sample using our NH$_4$Br carrier solution. To promote dissolution, we heated the spiked aliquots to 70 °C for 24-48 hours in a solution containing 4 mL of concentrated puriss HF and 6 mL 2M HNO$_3$ per gram of sample. After dissolution, we removed fluoride compounds by centrifuging. To precipitate Ag(Cl, Br), we added ~1 mL of 5% AgNO$_3$ to the supernatants in low light conditions and left the solutions to sit in a darkened room. After at least 12 hours, we recovered the Ag(Cl, Br) precipitates and transferred them and a small amount

of supernatant to centrifuge tubes. We vortexed the solutions, then let the precipitates settle for at least 5 minutes while periodically tapping the bases of the tubes on a table. Then, we centrifuged the tubes to collect the precipitates. All samples underwent two cycles of vortexing, settling, and centrifuging while still in acidic solutions, which facilitated scavenging and flocculation of colloidal Ag(Cl, Br). We then discarded the acidic supernatants and rinsed the precipitates with deionized water. With the Ag(Cl, Br) precipitates now in water, we repeated the vortexing, tapping, and centrifuging steps twice to fully rinse

the material and remove residual acids. After discarding the water, we dried the precipitates overnight at 70 °C. Finally, we packed the Ag(Cl, Br) precipitates into nitric acid cleaned stainless steel cathodes for AMS measurement of $^{35}Cl/^{37}Cl$ at CAMS.

We prepared aliquots for $^{35}Cl/^{37}Cl$ measurements in two analytical batches in July 2020 and February 2021, respectively. We used $^{35}Cl/^{37}Cl$ and batch-specific process blanks (Table 3) to calculate total sample chloride through standard isotope dilution methods (Faure and Mensing, 2005). Aliquots for the analytical batch prepared in July 2020 were not separated

from the full sample using a micro riffle splitter. The total Cl concentrations for these samples were corrected using process blank CLBLK-AQ6 (measured $^{35}Cl/^{37}Cl$ = 0.938 ± 0.0102). Aliquots for the batch prepared in February 2021 were separated from the full sample after homogenization with a micro riffle splitter. Total sample Cl concentrations for the February 2021 batch were corrected using process blank CLBLK-AQ8 (measured $^{35}Cl/^{37}Cl$ = 0.936 ± 0.0006).

### 2.2.3 Characterization of cosmogenic $^{36}Cl$: Updated workflow

We characterized cosmogenic $^{36}Cl$ concentrations for the updated workflow ("B") splits on Ag(Cl, Br) matrices prepared from ≤ 20 g of HNO$_3$-etched rock sample. We carried out the $^{36}Cl$ extractions in two analytical batches in August 2020 and May 2021. Because we determined the total sample chloride content prior to chemistry on the full sample, we were able to adjust the amount of rock sample and natural-ratio Weeks Island Halite carrier used for $^{36}Cl$ analyses to ensure consistent total Cl among all targets in each analytical batch while keeping expected $^{36}Cl/Cl$ for all samples well above

laboratory blank values (Table 4). For higher-Cl samples (YGT18-32B and YGT18-33B), no natural-ratio Cl carrier was needed, and optimal expected $^{36}Cl/Cl$ and amount of Cl in the Ag(Cl, Br) target (~750-1000 µg Cl) were achieved by adjusting the amount of rock digested. For low-Cl samples, optimal expected $^{36}Cl/Cl$ and total Cl were achieved by adding an appropriate amount of natural-ratio Weeks Island Halite carrier.

**Table 3: Laboratory information for test sample $^{35}Cl/^{37}Cl$ measurements and total Cl concentration determinations.**

| Sample ID | Sample mass (g) | Cl added from spike[a] (µg) | Memory-corrected $^{35}Cl/^{37}Cl$ | $^{35}Cl/^{37}Cl$ uncertainty | AMS uncertainty (%) | Blank-corrected total sample Cl (µg) | Sample Cl concentration (µg g$^{-1}$) | Sample Cl concentration uncertainty (µg g$^{-1}$) | Total Cl concentration measurement uncertainty (%) | Blank contribution to Cl concentration (%) |
|---|---|---|---|---|---|---|---|---|---|---|
| *Alaskan basalt samples* | | | | | | | | | | |
| 19SEAK-01A[b] | 20.1881 | 773 | 1.272 | 0.0132 | 1.04% | 292.1 | 14.47 | 0.17 | 1.19% | 1.47% |
| 19SEAK-01B-1[c] | 1.2063 | 79 | 1.133 | 0.0056 | 0.50% | 16.2 | 13.47 | 0.10 | 0.77% | 1.68% |
| 19SEAK-01B-2[d] | 1.1484 | 75 | 1.110 | 0.0004 | 0.04% | 13.7 | 11.94 | 0.04 | 0.37% | 2.01% |
| 19SEAK-02A[b] | 20.1015 | 774 | 1.104 | 0.0214 | 1.93% | 132.4 | 6.59 | 0.14 | 2.08% | 3.18% |
| 19SEAK-02B-1[c] | 1.2054 | 79 | 1.026 | 0.0136 | 1.33% | 6.9 | 5.75 | 0.09 | 1.51% | 3.85% |
| 19SEAK-02B-2[d] | 1.0383 | 76 | 1.014 | 0.0004 | 0.04% | 5.9 | 5.70 | 0.02 | 0.39% | 2.46% |
| 19SEAK-12A[b] | 20.0392 | 770 | 1.588 | 0.0265 | 1.67% | 683.0 | 34.08 | 0.60 | 1.77% | 0.63% |
| 19SEAK-12B-1[c] | 1.2252 | 79 | 1.347 | 0.0058 | 0.43% | 38.0 | 30.98 | 0.22 | 0.72% | 0.73% |
| 19SEAK-12B-2[d] | 1.0184 | 80 | 1.317 | 0.0005 | 0.04% | 35.5 | 35.03 | 0.13 | 0.36% | 0.42% |
| 19SEAK-13A[b] | 20.0511 | 775 | 1.579 | 0.0095 | 0.60% | 673.9 | 33.61 | 0.28 | 0.82% | 0.64% |
| 19SEAK-13B-1[c] | 1.2150 | 79 | 1.377 | 0.0059 | 0.43% | 41.6 | 34.27 | 0.25 | 0.72% | 0.66% |
| *Yellowstone rhyolite samples* | | | | | | | | | | |
| YGT18-31A[b] | 20.0445 | 774 | 1.672 | 0.0095 | 0.57% | 821.7 | 40.99 | 0.33 | 0.80% | 0.53% |
| YGT18-31B-1[c] | 1.2062 | 79 | 1.439 | 0.0062 | 0.43% | 49.2 | 40.76 | 0.29 | 0.72% | 0.56% |
| YGT18-31B-2[d] | 0.9412 | 80 | 1.362 | 0.0005 | 0.04% | 40.4 | 43.19 | 0.16 | 0.36% | 0.37% |
| YGT18-32A[b] | 20.0639 | 769 | 1.858 | 0.0101 | 0.54% | 1173.7 | 58.50 | 0.46 | 0.78% | 0.37% |
| YGT18-32B-1[c] | 1.2209 | 79 | 1.578 | 0.0068 | 0.43% | 68.4 | 55.98 | 0.40 | 0.72% | 0.40% |
| YGT18-33A[b] | 20.0619 | 770 | 1.804 | 0.0095 | 0.53% | 1060.1 | 52.84 | 0.41 | 0.77% | 0.41% |
| YGT18-33B-1[c] | 1.2086 | 79 | 1.528 | 0.0066 | 0.43% | 61.1 | 50.51 | 0.36 | 0.72% | 0.45% |
| *Process blanks* | | | | | | | | | | |
| CLBLK-23 | - | 774 | 0.940 | 0.0313 | 3.33% | - | - | - | - | - |
| CLBLK-AQ6 | - | 78 | 0.938 | 0.0102 | 1.09% | - | - | - | - | - |
| CLBLK-AQ8 | - | 77 | 0.936 | 0.0006 | 0.07% | - | - | - | - | - |

[a] All samples were spiked with "LLNL Spike A", which has a $^{35}Cl/^{37}Cl$ of 0.934 and a Cl concentration of 1285 ± 4 µg g$^{-1}$.

[b] Sample prepared using the standard workflow. Sample Cl concentration corrected using process blank CLBLK-23.

[c] Sample prepared using the updated workflow, without using a micro riffle splitter for homogenization. In addition to the $^{37}Cl$-enriched spike, sample received ~4000 µg Br from a ~10,000 µg g$^{-1}$ NH$_4$Br carrier solution. Sample Cl concentration corrected using process blank CLBLK-AQ6.

[d] Sample prepared using the updated workflow, with homogenization using a micro riffle splitter. In addition to the $^{37}Cl$-enriched spike, sample received ~4000 µg Br from a ~10,000 µg g$^{-1}$ NH4Br carrier solution. Sample Cl concentration corrected using process blank CLBLK-AQ8.

Each analytical batch contained a single process blank with Weeks Island Halite carrier and NH$_4$Br carrier. To account for the variable amounts of Weeks Island Halite carrier added to samples within a batch, the process blanks received the average amount of Weeks Island Halite carrier added to the samples in the batch (Table 4). No samples or blanks received $^{37}Cl$-enriched spike solution for the $^{36}Cl$ extraction. All samples and blanks received NH$_4$Br carrier, which served to increase the size of the final precipitate. We added enough NH$_4$Br carrier to each sample within an analytical batch such that the moles of Ag(Cl, Br) were equivalent to the moles of AgCl when precipitating 2000 µg of Cl. Because all samples in an analytical batch had a comparable amount of total Cl after the Weeks Island Halite carrier additions, each sample and process blank in a batch received the same amount of Br carrier.

After the Cl and Br carrier additions, we prepared the Ag(Cl, Br) targets using a modified version of the procedures outlined in Stone et al. (1996) and Licciardi et al. (2008). We dissolved the sample grains in a solution containing 4 mL of

concentrated puriss HF and 6 mL 2M $HNO_3$ per gram of sample. The solutions were heated to ~70 °C and left to dissolve over

240 two to three days. After dissolution, we removed insoluble fluorides through centrifuging. We transferred the supernatants into Teflon beakers and heated them to ~160 °C before adding ~1 mL 5% $AgNO_3$ under low light conditions. Samples were left to sit in a darkened cabinet for at least 12 hours to allow Ag(Cl, Br) to precipitate. We recovered the Ag(Cl, Br) precipitates and then dissolved them in 3.6M $NH_3$. To remove $^{36}S$, an interfering isobar of $^{36}Cl$, we separated Cl from S by the addition of saturated $BaNO_3$ to the sample solution. After two to three days, when $BaSO_4$ crystals had formed in all samples, we transferred

the clear supernatant to a new container. We precipitated Ag(Cl, Br) a final time by acidifying the solution with 2M $HNO_3$ and adding ~1 mL 5% $AgNO_3$. After letting the samples sit for at least 12 hours in a darkened cabinet, we recovered the Ag(Cl, Br), rinsed the precipitate in deionized water, and dried the material at 70 °C overnight. The precipitates were then sent to CAMS where they were loaded into stainless steel targets pre-packed with AgBr and analyzed for $^{36}Cl/Cl$.

**Table 4: Laboratory information for test sample $^{36}Cl$ concentration determinations.**

| Sample ID | Sample mass (g) | Sample Cl concentration (µg g⁻¹) | Cl added (µg) | Memory-corrected $^{36}Cl$/Cl | Ratio uncertainty | AMS uncertainty (%) | Blank-corrected $^{36}Cl$ concentration (atoms g⁻¹) | $^{36}Cl$ concentration uncertainty (atoms g⁻¹) | Total $^{36}Cl$ concentration measurement uncertainty (%) | Blank contribution to $^{36}Cl$ concentration (%) |
|---|---|---|---|---|---|---|---|---|---|---|
| *Alaskan basalt samples* | | | | | | | | | | |
| 19SEAK-01A[a, c] | 20.188 | 14.47 ± 0.17 | 773 | 6.89E-14 | 1.95E-15 | 2.83% | 1.08E+05 | 3.19E+03 | 2.96% | 3.19% |
| 19SEAK-01B[b, e] | 12.0014 | 12.71 ± 0.11[f] | 358 | 1.47E-13 | 3.39E-15 | 2.30% | 1.05E+05 | 2.45E+03 | 2.33% | 0.91% |
| 19SEAK-02A[a, c] | 20.102 | 6.59 ± 0.14 | 774 | 9.28E-14 | 2.17E-15 | 2.34% | 1.35E+05 | 3.27E+03 | 2.42% | 2.58% |
| 19SEAK-02B[b, e] | 11.5081 | 5.73 ± 0.09[f] | 435 | 1.74E-13 | 5.07E-15 | 2.92% | 1.28E+05 | 3.73E+03 | 2.93% | 0.78% |
| 19SEAK-12A[a, c] | 20.039 | 34.08 ± 0.60 | 770 | 2.51E-14 | 1.27E-15 | 5.07% | 4.54E+04 | 2.53E+03 | 5.57% | 7.31% |
| 19SEAK-12B[b, d] | 9.9516 | 33.0 ± 0.26[f] | 604 | 2.94E-14 | 1.63E-15 | 5.54% | 4.18E+04 | 2.68E+03 | 6.43% | 10.72% |
| 19SEAK-13A[a, c] | 20.051 | 33.61 ± 0.28 | 775 | 2.60E-14 | 1.09E-15 | 4.19% | 4.72E+04 | 2.18E+03 | 4.62% | 7.05% |
| 19SEAK-13B[b, d] | 16.9701 | 34.27 ± 0.25 | 308 | 5.81E-14 | 2.47E-15 | 4.25% | 4.87E+04 | 2.24E+03 | 4.59% | 5.69% |
| *Yellowstone rhyolite samples* | | | | | | | | | | |
| YGT18-31A[a, c] | 20.045 | 41.19 ± 0.33 | 774 | 3.01E-13 | 5.66E-15 | 1.88% | 6.22E+05 | 1.18E+04 | 1.89% | 0.57% |
| YGT18-31B[b, e] | 7.9558 | 41.97 ± 0.33[f] | 157 | 6.13E-13 | 1.31E-14 | 2.14% | 6.41E+05 | 1.37E+04 | 2.14% | 0.23% |
| YGT18-32A[a, c] | 20.064 | 58.50 ± 0.46 | 769 | 2.91E-13 | 5.46E-15 | 1.88% | 6.84E+05 | 1.29E+04 | 1.89% | 0.52% |
| YGT18-32B[b, d] | 14.973 | 55.98 ± 0.4 | 0 | 7.33E-13 | 1.21E-14 | 1.65% | 6.94E+05 | 1.15E+04 | 1.65% | 0.48% |
| YGT18-33A[a, c] | 20.062 | 52.84 ± 0.41 | 770 | 2.95E-13 | 5.55E-15 | 1.88% | 6.67E+05 | 1.26E+04 | 1.89% | 0.53% |
| YGT18-33B[b, d] | 16.9742 | 50.51 ± 0.36 | 0 | 7.74E-13 | 1.96E-14 | 2.54% | 6.61E+05 | 1.68E+04 | 2.54% | 0.10% |
| *Process blanks* | | | | | | | | | | |
| CLBLK-23[a] | - | | 774 | 2.59E-15 | 3.41E-16 | 13.17% | - | - | - | |
| CLBLK-25[b] | - | | 458 | 6.41E-15 | 1.95E-15 | 30.40% | - | - | - | |
| CLBLK-26[b] | - | | 287 | 2.37E-15 | 9.24E-16 | 38.99% | - | - | - | |

[a] Sample prepared using the standard workflow. Sample spiked with "LLNL Spike A", which has a $^{35}Cl/^{37}Cl$ of 0.93 and a Cl concentration of 1285 ± 4 µg g⁻¹.
[b] Sample prepared using the updated workflow. Sample received "Weeks Island Halite carrier" solution, which has a $^{35}Cl/^{37}Cl$ of 3.127 (i.e., natural ratio) and a Cl concentration of 1436 ± 9 µg g⁻¹. Sample also received ~4000 µg Br from a ~10,000 µg g⁻¹ $NH_4Br$ carrier solution.
[c] Sample $^{36}Cl$ concentration corrected using process blank CLBLK-23.
[d] Sample $^{36}Cl$ concentration corrected using process blank CLBLK-25.
[e] Sample $^{36}Cl$ concentration corrected using process blank CLBLK-26.
[f] Sample Cl concentration used for $^{36}Cl$ determinations is the average of two stable Cl aliquot measurements (one sample homogenized with a riffle splitter, and one sample with no homogenization; Table 3).

### 2.3.4 Isotopic analyses and process blank corrections for $^{36}$Cl concentrations

For both the standard method and updated workflow splits, $^{36}$Cl/Cl measurements were conducted at LLNL-CAMS. Sample ratios were normalized to KNSTD1600, which has a nominal $^{36}$Cl/$^{37}$Cl of $6.60 \times 10^{-12}$ and a $^{36}$Cl/Cl of $1.60 \times 10^{-12}$ (Sharma et al., 1990). To account for laboratory and AMS backgrounds, we corrected all sample $^{36}$Cl concentrations using ratios from batch-specific process blanks. The standard method ("A") splits were processed in a single analytical batch in November 2019 and their $^{36}$Cl concentrations were corrected using a batch-specific blank $^{36}$Cl/Cl of $2.59 \pm 0.341 \times 10^{-15}$ (CLBLK-23; equivalent to $7.18 \times 10^4$ $^{36}$Cl atoms). The process blank for this batch contained 774 µg of Cl from the $^{37}$Cl-enriched spike solution and no Br carrier (Table 4). The updated workflow ("B") splits were processed in two analytical batches in August 2020 and May 2021, respectively. The batch processed in August 2020 included four geologic samples and a blank (CLBLK-25) containing 458 µg of Cl from the natural Cl ratio Weeks Island Halite carrier and ~4000 µg of Br from the NH$_4$Br carrier. Sample $^{36}$Cl concentrations for the August 2020 batch were corrected using a process blank $^{36}$Cl/Cl of $6.41 \pm 1.95 \times 10^{-15}$ (equivalent to $4.99 \times 10^4$ $^{36}$Cl atoms). The batch processed in May 2021 included three geologic samples (Table 4) and a process blank (CLBLK-26) that contained 287 µg of Cl from the natural Cl ratio Weeks Island Halite carrier and ~4000 µg of Br from the NH$_4$Br carrier. Sample $^{36}$Cl concentrations for the May 2021 batch were corrected using a process blank $^{36}$Cl/Cl of $2.37 \pm 0.924 \times 10^{-15}$ (equivalent to $1.16 \times 10^4$ $^{36}$Cl atoms).

### 3 Results

### 3.1 Cl dilution series

Measurements of $^{35}$Cl/$^{37}$Cl from dilutions of the Wildcat spike and Weeks Island Halite carrier agree well with expected values over a range of 1.0 to 2.5 (Table 1; Fig. 3) with analytical uncertainties ranging between 0.0079% and 0.032%. The close agreement between the measured and expected $^{35}$Cl/$^{37}$Cl across the range of values demonstrates that samples analyzed on the low-energy end of the AMS yield accurate and highly precise stable Cl determinations. This result represents a marked improvement over past $^{35}$Cl/$^{37}$Cl measurement uncertainties on the high-energy end of the AMS, which average ~1%, and demonstrates that low energy $^{35}$Cl/$^{37}$Cl measurements are a favorable alternative to traditional methods.

Measurements of $^{35}$Cl/$^{37}$Cl from dilutions of the Wildcat spike and the NH$_4$Br carrier ranged from 1.012 to 1.036 (Table 2). Counting uncertainties for each target ranged between 0.0045% and 0.011%. Although we observe a general trend toward increasing measured $^{35}$Cl/$^{37}$Cl in dilutions with less $^{37}$Cl-enriched Wildcat spike, isotope dilution calculations reveal that all dilutions contain < 2.5 µg Cl per gram of NH$_4$Br (mean = $1.75 \pm 0.473$ µg g$^{-1}$); in other words, Cl contamination is negligible in our bromine carrier. Thus, the commercial NH$_4$Br source is suitable for the procedures presented here if samples contain at least 20 µg of total Cl, which is well below the typical target amount for whole rock and mineral separates analyzed for cosmogenic $^{36}$Cl exposure dating. For samples with total Cl masses < 20 µg, alternative preparation protocols that do not

use NH4Br carrier (e.g., Anderson et al., 2022) would be necessary. It should be noted, however, that those alternative protocols are not suitable for silicate materials that require digestion in HF during sample dissolution (*Section 4, Discussion, below*).

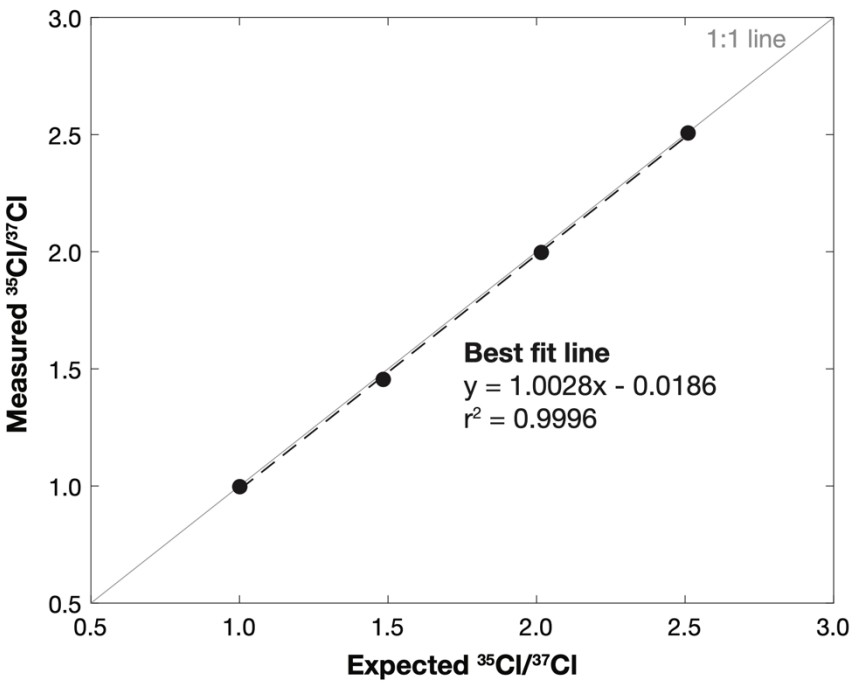

**Figure 3: Measured vs. expected $^{35}Cl/^{37}Cl$ for dilutions of the UNH Wildcat spike and the Weeks Island Halite carrier. Error bars on measured $^{35}Cl/^{37}Cl$ are too small to be visible on the figure. Solid gray diagonal line shows a 1:1 relationship between expected and measured $^{35}Cl/^{37}Cl$ and dotted black line shows the best fit line between the data points. The equation and $r^2$ value for the best fit line are also provided.**

## 3.2 Test samples

We successfully measured $^{35}Cl/^{37}Cl$ and $^{36}Cl/Cl$ on test samples at levels that were well above process blank values (Tables 3 and 4). For the "A" splits prepared using the standard workflow, errors on $^{35}Cl/^{37}Cl$ measurements ranged from 0.53% to 1.93%, resulting in total Cl concentration uncertainties (including process blank corrections) between 0.77% to 2.08%. Analytical uncertainty for $^{35}Cl/^{37}Cl$ analyses on the "B" splits prepared using the updated workflow presented here ranged from 0.04% to 1.33%, corresponding to total Cl concentration uncertainties (including process blank corrections) from 295 0.36% to 1.51%. These results demonstrate that pre-accelerator measurements of stable Cl isotope ratios can provide higher precision than measurements on the post-accelerator end of the AMS. Blank-corrected total Cl concentrations for test samples varied from ~6-35 µg g$^{-1}$ Cl for the Alaskan basalts to ~41-60 µg g$^{-1}$ Cl for the Yellowstone rhyolites. (Table 3; Fig. 4). Process blank contributions for samples prepared with both workflows are comparable (Table 3). Total Cl concentrations for the "A" 300 and "B" splits do not overlap at 2-sigma uncertainty for all samples (Table 3; Fig. 4). For the three Yellowstone rhyolite

samples (YGT18-31, YGT18-32, and YGT18-33), total Cl determinations for the "A" and "B" splits are within 5% of one another. For the four Alaskan basalt samples (19SEAK-01, 19SEAK-02, 19SEAK-12, and 19SEAK-13), the difference in total Cl concentration between the "A" and "B" splits ranges from 2% to 13%. This scatter is likely due to small-scale compositional heterogeneity in rock sample composition or dissolution.

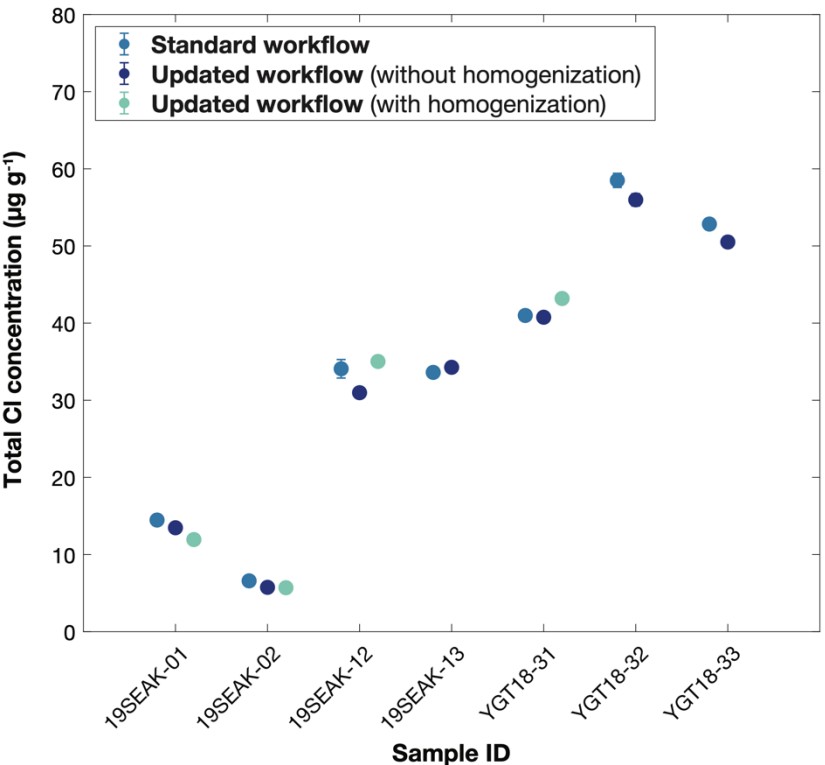

Figure 4: Comparison of total Cl concentrations (µg g⁻¹) for samples measured using the standard workflow "A" splits (blue dots; all samples), the updated workflow "B-1" splits with aliquots separated without homogenization (purple dots; all samples), and the updated workflow "B-2" splits with aliquots separated after homogenization with a riffle splitter (green dots; four samples only). With the exception of 19SEAK-12A and YGT18-32A, total uncertainties for each sample (Table 3) are smaller than the symbol size. For 19SEAK12-A and YGT18-32A, 2-sigma uncertainty is shown as a vertical line.

$^{36}$Cl concentrations for the standard method "A" splits ranged from $4.54 \times 10^4$ to $6.84 \times 10^5$ atoms g⁻¹, with total measurement uncertainties (including process blank corrections) between 1.89% and 5.57% (Table 4). For the updated workflow "B" splits, $^{36}$Cl concentrations ranged from $4.18 \times 10^4$ to $6.94 \times 10^5$ atoms g⁻¹; total measurement uncertainties ranged from 1.65% to 6.43%. Uncertainties on $^{36}$Cl/Cl measurements are higher than on $^{35}$Cl/$^{37}$Cl measurements, which is not surprising given the much lower ratios measured for $^{36}$Cl/Cl. $^{36}$Cl concentrations for all sample pairs agree within 1-sigma measurement uncertainty. There are no systematic variations in $^{36}$Cl concentration between the two preparation workflows (Fig. 5), and process blank contributions to $^{36}$Cl concentrations are also comparable for all samples (Table 4). This set of measurements demonstrates that the $^{36}$Cl analyses for each preparation method provide equivalent results within uncertainty.

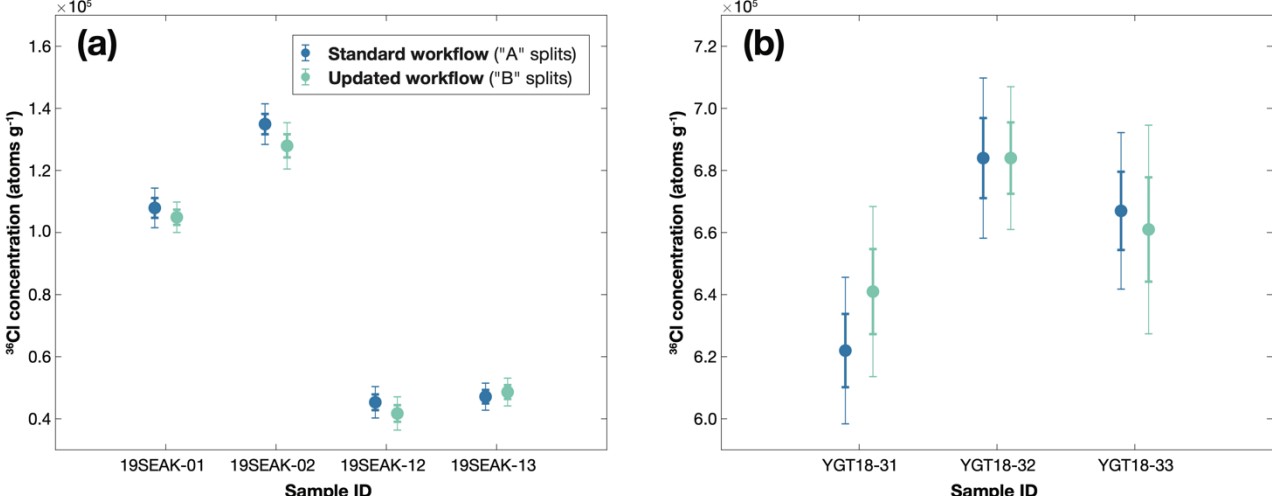

**Figure 5: Comparison of blank-corrected $^{36}$Cl concentrations (atoms g$^{-1}$) for the geologic test samples (a: Alaskan basalt; b:**
**Yellowstone rhyolite) prepared using the standard workflow ("A" splits; blue dots) and the updated workflow presented here ("B"**
**splits; green dots). Measurement uncertainties for each sample are shown at 1-sigma (thick vertical lines) and 2-sigma (thin vertical**
**lines). Although measurement uncertainties on $^{36}$Cl concentrations for the three YGT18 samples appear visually larger than those**
**of the 19SEAK samples, percent uncertainties on both sets of samples are comparable (Table 4). For ease of comparison, the range**
**of the y-axis in both panels is $1.4 \times 10^5$ atoms g$^{-1}$.**

## 4 Discussion

The cosmogenic chlorine workflow presented here offers substantial advantages over conventional protocols across all levels of analysis, spanning from laboratory preparation to AMS measurement. Measurement of $^{35}$Cl/$^{37}$Cl on the low-energy end of the AMS line with consistent and substantially smaller total Cl loads in all targets results in total uncertainties on $^{35}$Cl/$^{37}$Cl measurements of <1%, and often <0.5%. Rock sample chlorine concentrations can thus be determined with high precision, which will reduce external uncertainties on calculated surface exposure ages. By characterizing $^{35}$Cl/$^{37}$Cl and total Cl prior to $^{36}$Cl measurement, rock samples with high Cl concentrations can also be identified and screened at an earlier stage of analysis. This is important because high native Cl concentrations in rocks can result in exposure ages with comparably larger external uncertainties due to the greater uncertainty in the $^{36}$Cl production rate from thermal and epithermal neutron capture on $^{35}$Cl (Marrero et al., 2016). Depending on the desired exposure age precision, these high native Cl samples may be excluded from further analysis. Furthermore, the $^{35}$Cl/$^{37}$Cl aliquot method presented here uses only ~1 g of etched sample to obtain total Cl concentrations (Stone, 2001). However, we note that this method can be applied to substantially less material than 1 g depending on a sample's total Cl concentration, aliquot homogeneity, and the need for high precision in the Cl concentration measurement. We chose to use 1 g of etched sample to help ensure that the stable Cl aliquot was representative of the whole sample, and because we suspected some samples could have exceptionally low Cl concentrations such that 1 g would be large enough to characterize Cl at levels where the $^{35}$Cl(n,$\gamma$)$^{36}$Cl reaction becomes a significant contributor to total

[36]Cl production. For the method presented here, a sample size equivalent to 5 μg Cl is likely sufficient, indicating that many of our samples could have been analyzed at high precision using only ~0.1 g of material. This is substantially less than the amount required for total Cl determinations by other techniques such as XRF, which can consume >10 g of material. The small amount of material required for total Cl measurements is thus a key advantage when sample sizes are limited (e.g., when analyzing mineral separates). Finally, and perhaps most importantly for cosmogenic chlorine extraction laboratories, our workflow reduces the use of isotopically enriched chlorine spike by up to 95% compared to conventional methods, which will considerably extend the lifespan of existing laboratory supplies.

While our laboratory protocols offer an improvement over conventional cosmogenic chlorine preparation methods, there is room for further refinement. For example, Anderson et al. (2022) presented an innovative approach wherein AgCl targets were prepared within a niobium matrix, resulting in successful measurement of $^{35}Cl/^{37}Cl$ on targets with Cl masses as low as 1 μg; they also observed a substantially reduced source memory compared to targets bulked with AgBr. However, a key distinction between their methodology and the one outlined here is that, unlike our silicate samples, Anderson et al.'s aqueous Cl samples did not require the use of HF in the dissolution process. Given niobium's solubility in HF acid, employing this method on silicate samples requires thorough rinsing to eliminate residual acid from AgCl precipitates before introducing niobium powder. HF remaining in the AgCl precipitates can also react to form fluoride compounds with an atomic mass of 35 (e.g., $^{16}O^{19}F$), which can interfere with measurement of $^{35}Cl/^{37}Cl$. Thus, removing all HF before adding niobium to AgCl precipitates is critical. On the other hand, AgCl is mildly water soluble (~2 μg mL$^{-1}$), and extensive rinsing may reduce the AgCl yield. Therefore, the challenge is to rinse samples thoroughly enough to remove all traces of HF while still maintaining a high AgCl yield. We have attempted this process with silicate-derived AgCl precipitates that contain ~50 μg of total Cl, but consistently and accurately measuring $^{35}Cl/^{37}Cl$ on such samples has proven difficult, likely due to both yield loss during the procedure and residual HF remaining in some targets even after thorough rinsing. Refining these procedures is the focus of ongoing experimentation. Nevertheless, our findings unequivocally show that $^{35}Cl/^{37}Cl$ on silicate samples can be measured with high precision on Ag(Cl, Br) when the total chlorine mass in each target exceeds 50 μg. Thus, while there are possibilities for improvement, we are confident that the procedures presented herein can be deployed with little modification in cosmogenic chlorine laboratories and AMS facilities.

We note that adopting these procedures would slightly increase the time and lab resources required to extract $^{36}Cl$ from rock samples. For example, although $^{35}Cl/^{37}Cl$ can be measured on the low-energy end of the AMS line, which does not require a "full-scale" AMS run, sample shipment and data processing would add some time between sample preparation and the receipt of final $^{36}Cl$ concentrations. The separate $^{35}Cl/^{37}Cl$ extraction adds approximately three to four days of laboratory work to the $^{36}Cl$ extraction timeline and also requires small amounts of reagents such as HF, HNO$_3$, and AgNO$_3$. However, in cases where total sample Cl is high, $^{36}Cl$ extractions can be carried out on smaller masses of milled rock, and thus the total volume of acid used in both the $^{35}Cl/^{37}Cl$ and $^{36}Cl$ extractions may be smaller than what was required to dissolve ~20 g of milled rock using the standard workflow. Furthermore, having knowledge of Cl concentration in advance of the $^{36}Cl$ analysis allows for screening out of samples with high Cl content that may be undesirable for certain applications. In such cases, this

workflow saves both time and resources that would otherwise be wasted on less useful $^{36}$Cl data. In terms of the analytical cost, presently at the CAMS facility there is no cost difference between analysing $^{35}$Cl/$^{37}$Cl on a separate, earlier target, and analysing both $^{35}$Cl/$^{37}$Cl and $^{36}$Cl/Cl on the same target during a single AMS run. However, for a $^{35}$Cl/$^{37}$Cl measurement used to screen samples (that is, when no later $^{36}$Cl/Cl analysis is performed) the analysis cost is ~1/7$^{th}$ that of the $^{36}$Cl/Cl inclusive analysis such that significant cost savings can be achieved when sample screening is prudent.

To implement the preparation and measurement workflow described above, we suggest three key practices. First, we recommend using a micro riffle splitter to separate subsamples for $^{35}$Cl/$^{37}$Cl, $^{36}$Cl/Cl, and major element analyses from the full rock sample. The homogenization step does not appear to be strictly necessary for the two lithologies we tested here (basalt and rhyolite; Fig. 4). We speculate that these fine-grained igneous rocks are already quite compositionally homogenous due to the scarcity or lack of phenocrysts and the high percentage of groundmass, and the milled grains therefore do not separate

much by composition after milling and during storage in plastic bags. However, we do encourage the use of a micro riffle splitter or other homogenization method (e.g., the "cone and quarter" technique) for coarse grained lithologies that may contain monomineralic grains when crushed to the 250-125 μm size fraction we recommend here, as we suspect these rocks may be more susceptible to separation by composition during storage. Second, we stress the importance of exercising caution when cleaning stainless steel AMS cathodes prior to sample loading. Early experiments indicated that the commercial laboratory

soaps we used to clean our AMS cathodes contain chlorine that is not removed from cathode surfaces by thorough rinsing with deionized water. Therefore, we suggest that laboratory users soak AMS cathodes in a weak (~1%) nitric acid solution after cleaning with laboratory soap. This step is essential to eliminate residual natural-ratio chlorine that may contaminate the cathode surfaces and erroneously raise measured $^{35}$Cl/$^{37}$Cl on spiked samples. Third, if samples are bulked with NH$_4$Br carrier, we recommend that laboratory users prepare "matrix blanks" of AgBr (with no added Cl) for measurement, which will allow

for accurate ion source memory corrections for unknown samples. Consistency in target matrices across a stable Cl analysis minimizes source memory and improves ion beam stability, contributing to more accurate and reproducible experimental outcomes.

     The workflow presented here uses a $^{37}$Cl-enriched spike solution for the $^{35}$Cl/$^{37}$Cl measurements, but it is also suitable for use with a $^{35}$Cl-enriched spike so long as the solution is sufficiently enriched relative to the natural $^{35}$Cl/$^{37}$Cl of 3.127. In

general, solutions enriched in $^{35}$Cl are more readily available than $^{37}$Cl-enriched solutions, so this may be an attractive alternative for laboratories that cannot acquire a $^{37}$Cl-enriched spike.

## 5 Conclusions

     Our workflow for extracting and measuring chlorine in silicate rocks improves upon standard preparation methods in several key ways. After crushing the rock and cleaning the mineral surfaces with dilute HNO$_3$, we characterize stable Cl ratios

on an up to ~1 g aliquot of rock removed from the full sample. $^{35}$Cl/$^{37}$Cl is then measured on the low-energy beam line of the AMS accelerator, allowing us to quickly determine total chlorine loads while minimizing source memory and reducing the

amount of $^{37}$Cl-enriched carrier solution used per sample by up to 95% compared to traditional methods. With $^{35}$Cl/$^{37}$Cl data in hand, we then extract Cl from rock samples for $^{36}$Cl/Cl measurements and bulk the AMS target material with bromine and/or natural-ratio chlorine solutions, without adding $^{37}$Cl-enriched spike. Experiments on seven geologic test samples reveal that the workflow presented here yields comparable or, in the case of the $^{35}$Cl/$^{37}$Cl measurements, improved results over the traditional workflow. Most notably, by measuring $^{35}$Cl/$^{37}$Cl on a ~1 g aliquot rather than a 20 g sample, the updated preparation methods use substantially less isotopically enriched spike solution than standard methods (~50-75 µg Cl versus ~750-1000 µg Cl). With lowered spike solution requirements, researchers can analyze many more samples using their remaining laboratory resources. Chlorine extraction laboratories will also be able to maintain control over the total chlorine content within and across analytical batches. Finally, in comparison to the standard $^{36}$Cl workflow, our method can identify samples with elevated native Cl concentrations at earlier stage of laboratory work, which can help researchers determine which of their rock samples should be prioritized for $^{36}$Cl analyses. Our hope is that these procedures will supplement existing laboratory and AMS workflows for cosmogenic Cl and enhance the effectiveness of $^{36}$Cl dating for a variety of geologic applications.

**Data availability**

All data are presented in Tables 1-4 of this technical note.

**Author contribution**

AJL and JML carried out the laboratory work. AJH and TSA made the $^{36}$Cl/Cl and $^{35}$Cl/$^{37}$Cl measurements. AJL wrote the first draft of the manuscript, and all authors contributed to experimental design, data analysis, and manuscript review and editing.

**Competing interests**

The authors declare that they have no competing interests.

## Acknowledgements

The authors thank Shasta Marrero and Irene Schimmelpfennig for their thoughtful and constructive reviews. We also thank John Stone and Keith Fifield for helpful discussions. AJL acknowledges funding from Queens College. Prepared in part by LLNL under Contract DE-AC52-07NA27344. This is LLNL-JRNL-861073.

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
