# Peer review of "Technical note: Optimizing the in situ cosmogenic 36Cl extraction and measurement workflow for geologic applications"

_EGUsphere, 2024_

## Author Comment (AC1)

**Response to reviewer comments**
Alia J. Lesnek, Joseph M. Licciardi, Alan J. Hidy, and Tyler S. Anderson

We would like to thank Shasta Marrero and Irene Schimmelpfennig for their thoughtful and supportive comments on our preprint. Below, we provide point-by-point responses to both reviewers' comments. The full original reviews are shown in black text, and our responses are in blue text. In many of our responses, we include quotes from the revised manuscript that demonstrate how we addressed the reviewers' comments. Line numbers with these quotes indicate the location of the text in the revised manuscript, with the additions to the manuscript in **bold blue text**.
* * *
**Review #1: Shasta Marrero**

**General Comments**
I'm very excited to see work on improving cosmogenic $^{36}Cl$ processing methods. This paper clearly presents a method for cosmogenic chlorine sample processing that splits the measurement of stable Cl from the cosmogenic measurement to improve various aspects of sample processing. There are a number of key concerns in the community around chlorine sample processing and this paper is a step towards solving some of these. Overall, the authors did an excellent job of communicating everything clearly and the paper was easy to follow. I have a couple of key points, but overall I recommend this paper for publication after minor revisions. This is a much-needed step forward.

Thank you.

**Specific Comments**
**(1)** I think the most important point of this paper is the ability to use significantly less isotopically enriched carrier (spike) for each chlorine sample. The importance of this point in this field cannot be overstated. Spike price has been increasing and there is the possibility of increasingly dwindling supplies in the future since it is not a commonly created reagent. This issue has come up regularly at conferences and is a concern for everyone in the field. The reduction of the carrier needed for each sample offers a significant improvement. This point could be highlighted even more, if desired.

We agree that this is one of the most important aspects of our paper. We have added further emphasis to this point in the introduction and conclusion sections. The relevant text is copied below:

Lines 97-99: **A key finding of our experiments is that compared to standard methods, our workflow reduces the use of costly isotopically enriched Cl spike solution by up to 95%,** which should increase the accessibility of $^{36}Cl$ dating for geologic applications, as laboratory users can prepare more samples with their existing supplies.

Lines 405-414: **Our workflow for extracting and measuring chlorine in silicate rocks improves upon standard preparation methods in several key ways.** After crushing the rock and cleaning the mineral surfaces with dilute $HNO^3$, we characterize stable Cl ratios on an up to ~1 g aliquot of rock removed from the full sample. $^{35}Cl/^{37}Cl$ is then measured on the low-energy beam line of the AMS accelerator, allowing us to quickly determine total chlorine loads while minimizing source memory and **reducing the amount of $^{37}Cl$-enriched carrier solution used per sample by up to 95% compared to traditional methods**. With $^{35}Cl/^{37}Cl$ data in hand, we then extract Cl from rock samples for $^{36}Cl/Cl$ measurements and bulk the AMS target material with bromine and/or natural-ratio chlorine solutions, without adding $^{37}Cl$-enriched spike.

Experiments on seven geologic test samples reveal that the workflow presented here yields comparable or, in the case of the $^{35}Cl/^{37}Cl$ measurements, improved results over the traditional workflow. **Most notably, by measuring $^{35}Cl/^{37}Cl$ on a ~1 g aliquot rather than a 20 g sample, the new preparation methods use substantially less isotopically enriched spike solution than standard methods** (~50-75 µg Cl versus ~750-1000 µg Cl). With lowered spike solution requirements, researchers can analyze many more samples using their remaining laboratory resources.

**(2)** The paper discusses only using $^{37}Cl$-enriched carrier and explains why this is preferable to the $^{35}Cl$-enriched carrier. However, many labs are using $^{35}Cl$-enriched carrier instead for a number of reasons (cheaper, more readily available), so I feel that this technique could be applicable to a broader range of labs more quickly if you address the issue of whether or not this technique would be possible with $^{35}Cl$-enriched carrier instead and if any changes might be needed (or when to proceed with caution). Obviously, this is not the ideal situation, but given how many labs use this carrier already, it would be nice to see it addressed directly.

This is a great point. We have added the following text to the revised manuscript:

Lines 68-70: Figure 1: Schematic of the standard workflow for cosmogenic $^{36}Cl$ analysis based on Licciardi et al. (2008). Black arrows indicate the order of steps for each stage of the process. **Note that while we use a $^{37}Cl$-enriched spike solution for the stable Cl extraction, these procedures are also suitable for use with a $^{35}Cl$-enriched spike.**

Lines 190-193: Figure 2: Schematic of the new workflow for cosmogenic $^{36}Cl$ analysis presented here. Black arrows indicate the order of steps for each stage of the process. **Note that while we use a $^{37}Cl$-enriched spike solution for the stable Cl extraction, these procedures are also suitable for use with a $^{35}Cl$-enriched spike.**

Lines 398-401:  **The workflow presented here uses a $^{37}Cl$-enriched spike solution for the $^{35}Cl/^{37}Cl$ measurements, but it is also suitable for use with a $^{35}Cl$-enriched spike so long as the solution is sufficiently enriched relative to the natural $^{35}Cl/^{37}Cl$ of 3.127.  In general, solutions enriched in $^{35}Cl$ are more readily available than $^{37}Cl$-enriched solutions, so this may be an attractive alternative for laboratories that cannot acquire a $^{37}Cl$-enriched spike.**

**(3)** This is a very practical paper, so I will bring up one more practical point: adding additional steps at the AMS facility can be challenging when you do not have an AMS nearby. I love the fact that measuring Cl in advance would help processing be more exact, but it will also add time to the processing as well as an extra trip to the accelerator which will add considerable time to each sample batch (and potentially more cost for additional measurements?), although this will affect some labs more than others (e.g. those where samples are delivered in infrequent large batches). This is not a huge point and I think the advantages are worth the additional time in this case, but maybe worth mentioning?

This is also a good point that both reviewers mentioned. As for the potential costs, CAMS updates analytical costs each fiscal year, so we are hesitant to include dollar amounts for AMS measurements in the manuscript. We are also not sure what other AMS facilities would charge if they implemented this workflow. That said, at CAMS, the separate $^{35}Cl/^{37}Cl$ analyses are included in the cost of a $^{36}Cl/Cl$ measurement. Thus, there is no analytical cost difference between the traditional workflow measuring both ratios on the same target and the workflow described here where $^{35}Cl/^{37}Cl$ is measured earlier on a separate target. However, for samples where only $^{35}Cl/^{37}Cl$ is measured, and $^{36}Cl$ not analyzed, the cost is

about ~1/7 of a $^{36}Cl/Cl$ measurement. We now address these issues toward the end of the discussion section:

Lines 366-379: **We note that adopting these procedures would slightly increase the time and lab resources required to extract $^{36}Cl$ from rock samples. For example, although $^{35}Cl/^{37}Cl$ can be measured on the low-energy end of the AMS line, which does not require a "full-scale" AMS run, sample shipment and data processing would add some time between sample preparation and the receipt of final $^{36}Cl$ concentrations. The separate $^{35}Cl/^{37}Cl$ extraction adds approximately three to four days of laboratory work to the $^{36}Cl$ extraction timeline and also requires small amounts of reagents such as HF, $HNO_3$, and $AgNO_3$. However, in cases where total sample Cl is high, $^{36}Cl$ extractions can be carried out on smaller masses of milled rock, and thus the total volume of acid used in both the $^{35}Cl/^{37}Cl$ and $^{36}Cl$ extractions may be smaller than what was required to dissolve ~20 g of milled rock using the standard workflow. Furthermore, having knowledge of Cl concentration in advance of the $^{36}Cl$ analysis allows for screening out of samples with high Cl content that may be undesirable for certain applications. In such cases, this workflow saves both time and resources that would otherwise be wasted on less useful $^{36}Cl$ data. In terms of the analytical cost, presently at the CAMS facility there is no cost difference between analysing $^{35}Cl/^{37}Cl$ on a separate, earlier target, and analysing both $^{35}Cl/^{37}Cl$ and $^{36}Cl/Cl$ on the same target during a single AMS run. However, for a $^{35}Cl/^{37}Cl$ measurement used to screen samples (that is, when no later $^{36}Cl/Cl$ analysis is performed) the analysis cost is ~1/7[th] that of the $^{36}Cl/Cl$ inclusive analysis such that significant cost savings can be achieved when sample screening is prudent.**

Here are a number of smaller points/questions:
- Are you using a syringe filter after the Ba step?
  We are not using a syringe filter.

- When crushing samples, why only use the 125-250 micron sized fraction for the bulk composition? There is the potential for some fractions to crush more easily than others, which would differentiate the different grain size fractions in terms of composition. Perhaps not a problem with your mostly-homogeneous basalts in this case, but perhaps for others?
  This is an interesting point. We are not aware of any differences in bulk composition between different crushed size fractions isolated from the same rock, although we have not tested this explicitly for different lithologies, and to our knowledge this has not been demonstrated by other research groups. We use the 125-250 micron sized fraction for the bulk composition in part for practical reasons because we routinely use this same grain size fraction for $^{36}Cl$ extraction from the full sample, and this therefore streamlines the milling steps for physical sample processing.

- Spoon splitting? As you state, this will matter more for some types of samples than others. Did you mix up the sample before dipping in the spoon? There are also other variations of splitting methods (such as cone and quarter) that are between a grab sample with a spoon and a riffle splitter that might preserve the idea of a more homogenous sample with less cleaning/concern (when it doesn't matter as much).
  These are good points. We have added the following to the discussion:

  Lines 385-388: However, we do encourage the use of a micro riffle splitter **or other homogenization method (e.g., the "cone and quarter" technique)** for coarse grained lithologies

that may contain monomineralic grains when crushed to the 250-125 μm size fraction we recommend here, as we suspect these rocks may be more susceptible to separation by composition during storage.

- Can you reuse the cathodes at all?
  At CAMS, the cathodes cannot be reused after they are taken out of the cesium source after measurement and exposed to air.

- Line 333: (e.g., when analyzing feldspar mineral separates) – remove the word "feldspar". No need to specify since any kind of mineral separate will generally be smaller amounts.
  Done.

- Line 337 - I really appreciate the discussion of other (as-of-yet unsuccessful) methods. I wish more papers had this.
  Thank you! We included this paragraph in case other labs wanted to try the Nb method with silicate samples. If anyone can do this successfully, we would love to hear about it.

**One additional note:** John Stone has been using a split procedure (measuring Cl content in one sample and $^{36}$Cl ratio in the other) for decades (mostly on mineral separates) so splitting these measurements is not necessarily a 'new' idea on its own. However, this paper does seem to be the first time that it has been quantitatively and publicly compared against other common procedures and provided in a level of detail that would allow/encourage others to use this method.

We thank both reviewers for bringing up this important point, as it was a much-discussed topic among the authors as we were conducting our experiments and writing the manuscript. We came to a similar conclusion as Shasta Marrero did above: while several labs have been using similar methods for quite some time, it would be beneficial to the cosmogenic $^{36}$Cl community to have a detailed and publicly available description of this workflow and its advantages. Since we knew that other researchers have been using similar techniques in their labs, we consciously avoided describing the workflow presented in our manuscript as a 'new method.' That being said, we could have done a better job of acknowledging the prior work of others, especially John Stone and Keith Fifield, in our paper. Although we were not in touch with them before our initial submission, we have since connected to discuss the similarities and differences between our approaches. In the revised version of the manuscript, we have added the following:

Lines 91-93: **This workflow is similar to procedures developed at the University of Washington Cosmogenic Isotope Laboratory (Stone et al., 1996; Stone, 2001), but with a key difference being that we strongly encourage laboratory users to prepare subsamples for $^{35}$Cl/$^{37}$Cl measurements in advance of $^{36}$Cl/Cl analyses.**

Lines 431-432: **We also thank John Stone and Keith Fifield for helpful discussions.**

Additionally, the proof of concept on whole-rock samples and many of the other processing steps (including explanations) are fantastic contributions.

Thanks again. We appreciate the support.

**Review #2: Irene Schimmelpfennig**

This technical note describes a non-standard sample preparation and measurement method for cosmogenic $^{36}$Cl determinations by AMS. This method is highly useful for measuring Cl more precisely and saving significant amounts of the isotopically enriched chlorine spike that is typically used for isotope dilution $^{36}$Cl AMS measurements, but becomes rare and expensive. Comparison of samples treated with the standard and the non-standard methods convincingly validate the non-routine protocol. This work is therefore very helpful and might be beneficial for other $^{36}$Cl users and AMS facilities. The manuscript is very easy to read and understand, and the figures are of high quality. I list a few comments and questions below, but can highly recommend the publication of this paper.

Thank you very much.

A significant drawback of the presented non-standard method is that the sample preparation workload and waiting time for the final AMS results is close to doubled. Thus, the main argument of reducing the expensive spike might cancel out by the costs for additional work time and the additional AMS measurement. In my opinion, this should be included in the discussion, as it is probably the main reason why this method has not been adapted by more users so far.

Thanks for bringing this up. We agree that this point should be presented in our paper. Please see our response to Shasta Marrero's Specific Comment 3.

In this context I was wondering if a comprise could consist in combining stages 2 and 3 of the new workflow by spiking the sample grains for $^{36}$Cl measurements, as in the standard method, but with less spike and by bulking with the Br carrier, as in the non-standard method. I guess this should be possible at least for low-Cl samples (for higher-Cl samples the isotope dilution would not be sufficient), e.g. after a preliminary determination of approximate Cl concentrations for a set of samples from the same location. Has this been considered or tested?

This could certainly be done if the enriched spike is $^{36}$Cl-free, though it is not recommended otherwise. While not considered for this particular workflow, bulking with Br for the $^{36}$Cl sample is something that is done commonly at CAMS when $^{36}$Cl/Cl is expected to be low and there is a desire to not dilute the ratio further with carrier addition. When feasible, this could be a partial solution to the longer processing time with the split sample workflow, so long as approximate total Cl concentrations can be reliably estimated. However, if you do this and your estimates are off for some samples, you may end up with a larger range of total Cl loads across an analytical batch, or even run the risk of not diluting the Cl isotopes sufficiently. This may be worth the potential tradeoff if you need results quickly. You would still be using a smaller amount of enriched Cl spike, which is an improvement over the "standard" workflow.

Throughout the manuscript, the minimum $^{36}$Cl/Cl ratios necessary for precise $^{36}$Cl concentration determinations was not given much importance. I think this aspect could be more considered.

Thanks for the suggestion to clarify this. In our workflow, we are aiming to balance the sample mass, spike/carrier additions, and estimated Cl ratios such that each sample contains about the same amount of total Cl as the others in the analytical batch (between 500-1000 ug Cl, either from the sample only, or from the sample plus the natural Cl spike) while keeping the expected $^{36}$Cl/Cl ratios well above lab blank values (at UNH, these are usually in the 1e-15 to 7e-15 range). We have added the following text to address this point:

Lines 217-224: Because we determined the total sample chloride content prior to chemistry on the full sample, we were able to adjust the amount of rock sample and natural-ratio Weeks Island Halite carrier used for $^{36}$Cl analyses to ensure consistent total Cl among all targets in each analytical batch **while keeping expected $^{36}$Cl/Cl for all samples well above laboratory blank values** (Table 4). For higher-Cl samples (YGT18-32B and YGT18-33B), no natural-ratio Cl carrier was needed, **and optimal expected $^{36}$Cl/Cl** and amount of Cl in the Ag(Cl, Br) target (~750-1000 µg Cl) were achieved by adjusting the amount of rock digested. For low-Cl samples, **optimal expected $^{36}$Cl/Cl** and total Cl were achieved by adding an appropriate amount of natural-ratio Weeks Island Halite carrier.

- line 55: "A consistent sample mass (usually ~10-20 g of milled rock for whole-rock silicates or ~5-10 g of isolated mineral separates)": Theses masses should be adjusted based on age estimates, altitude and compositions of the samples and might need to be substantially higher to obtain $^{36}$Cl/Cl ratios significantly above the blanc, e.g. for very young surfaces (see e.g. our $^{36}$Cl dating of Last Glacial and last-millennium glacial surfaces in Charton et al., 2022 https://doi.org/10.1016/j.quascirev.2022.107461 ).
  Very true. We have amended the text to emphasize this point:

  Lines 55-60: In situ $^{36}$Cl concentrations are typically measured via AMS methods on targets prepared in an AgCl matrix (Fig. 1; Licciardi et al., 2008). **To ensure that Cl isotope ratios are well above laboratory blank values, consistent sample masses are prepared for Cl isotope analysis; depending on anticipated $^{36}$Cl inventories (which are a function of exposure duration, altitude, and sample composition), each sample usually consists of ~10-20 g of milled rock for whole-rock silicates or ~5-10 g of isolated mineral separates. Rock samples are** spiked with isotopically enriched Cl carrier solution such that total sample Cl (from $^{35}$Cl/$^{37}$Cl) and $^{36}$Cl concentrations (from $^{36}$Cl/$^{37}$Cl or $^{36}$Cl/Cl) can be determined through isotope dilution methods (Faure and Mensing, 2005).

- Lines 211-212: the amount of rock sample is said to be adjusted "to ensure consistent total Cl among all targets in each analytical batch". Is the impact of the sample mass on the $^{36}$Cl/Cl neglected in this calculation? I guess that optimizing amounts of Cl and $^{36}$Cl can be in conflict for some low-$^{36}$Cl samples? In this context it would be helpful to have a better idea of the $^{36}$Cl blank contributions in the presented sample $^{36}$Cl determinations. According to my back-of-the-envelope estimates they are on the order of 1-6%. (BTW, this is also missing for the Cl determinations.)
  We agree with your first point; please see our response above. To your second point, we have now added columns to tables 3 and 4 with our process blank uncertainty contributions for each sample.

- Line 282 (and complementing my previous questions): it would be interesting to know if Cl and $^{36}$Cl blank corrections are generally lower, similar or higher with the non-standard method.
  For the samples we measured, the blank corrections are similar between the two workflows. See revised tables 3-4. We also added the following text:

  Lines 298-299: **Process blank contributions for samples prepared with both workflows are comparable (Table 3).**

Lines 315-316: There are no systematic variations in $^{36}$Cl concentration between the two preparation workflows (Fig. 5), **and process blank contributions to $^{36}$Cl concentrations are also comparable for all samples (Table 4).**

- Line 295: I might have missed it, but couldn't find the explanation why the uncertainties in the concentrations of the homogenized samples are that much smaller than in the non-homogenized samples. Please clarify if not done yet.
  After reviewing the data, we realized that the difference in total uncertainty shown in the previous version of the manuscript was due to an error in our previous uncertainty calculations. We have updated Tables 3-4, Figures 4-5, and the corresponding text with new values for the total measurement uncertainties, and there is now no major difference between the Cl concentration uncertainties between the homogenized and non-homogenized samples.

  Lines 291-304: We successfully measured $^{35}$Cl/$^{37}$Cl and $^{36}$Cl/Cl on test samples at levels that were well above process blank values (Tables 3 and 4). For the "A" splits prepared using the standard workflow, errors on $^{35}$Cl/$^{37}$Cl measurements ranged from 0.53% to 1.93%, **resulting in total Cl concentration uncertainties (including process blank corrections) between 0.77% to 2.08%. Analytical uncertainty** for $^{35}$Cl/$^{37}$Cl analyses on the "B" splits prepared using the new workflow presented here ranged **from 0.04% to 1.33%, corresponding to total Cl concentration uncertainties (including process blank corrections) from 0.36% to 1.51%. These results demonstrate** that pre-accelerator measurements of stable Cl isotope ratios can provide higher precision than measurements on the post-accelerator end of the AMS. Blank-corrected total Cl concentrations for test samples varied from ~6-**35** μg g$^{-1}$ Cl for the Alaskan basalts to ~**41**-60 μg g$^{-1}$ Cl for the Yellowstone rhyolites. (Table 3; Fig. 4). **Process blank contributions for samples prepared with both workflows are comparable (Table 3). Total Cl concentrations for the "A" and "B" splits do not overlap at 2-sigma uncertainty for all samples (Table 3; Fig. 4). For the three Yellowstone rhyolite samples (YGT18-31, YGT18-32, and YGT18-33), total Cl determinations for the "A" and "B" splits are within 5% of one another. For the four Alaskan basalt samples (19SEAK-01, 19SEAK-02, 19SEAK-12, and 19SEAK-13), the difference in total Cl concentration between the "A" and "B" splits ranges from 2% to 13%. This scatter is likely due to small-scale compositional heterogeneity in rock sample composition or dissolution.**

[Figure]

Lines 205-309: Figure 4: Comparison of total Cl concentrations (µg g$^{-1}$) for samples measured using the standard workflow "A" splits (blue dots; all samples), the new workflow **"B-1"** splits with aliquots separated without homogenization (purple dots; all samples), and the new workflow **"B-2"** splits with aliquots separated after homogenization with a riffle splitter (green dots; four samples only). **With the exception of 19SEAK-12A, total uncertainties for each sample (Table 3) are smaller than the symbol size. For 19SEAK12-A, 2-sigma uncertainty is shown as a vertical line.**

- Lines 318-219: "Rock sample chlorine concentrations can thus be determined with high precision". It might be interesting to add a sentence about how this can impact the precision of the final $^{36}$Cl concentrations.
  We have added the following to the revised text:

  Line 330: Rock sample chlorine concentrations can thus be determined with high precision, **which will reduce external uncertainties on calculated surface exposure ages.**

Finally, I was surprised that John Stone and Keith Fifield are not mentioned, as they have been using this method for many years and have willingly been sharing detailed information about it. If this work benefitted from their experience, I guess it would be appropriate to acknowledge them.

Thank you for pointing this out. Please see our response to Shasta Marrero's "one additional note" above, which addresses this topic.